# Mitochondrial Metabolic Signatures in Hepatocellular Carcinoma

**DOI:** 10.3390/cells10081901

**Published:** 2021-07-27

**Authors:** Ho-Yeop Lee, Ha Thi Nga, Jingwen Tian, Hyon-Seung Yi

**Affiliations:** 1Laboratory of Endocrinology and Immune System, Chungnam National University School of Medicine, Daejeon 35015, Korea; leehy0280@o.cnu.ac.kr (H.-Y.L.); ngahahus@o.cnu.ac.kr (H.T.N.); jingwen0729@o.cnu.ac.kr (J.T.); 2Department of Medical Science, Chungnam National University School of Medicine, Daejeon 35015, Korea; 3Research Center for Endocrine and Metabolic Diseases, Chungnam National University School of Medicine, Daejeon 35015, Korea

**Keywords:** hepatocellular carcinoma, mitochondria, mitochondrial unfolded protein response, glycolysis, mitoribosome

## Abstract

Hepatocellular carcinoma (HCC) is one of the leading causes of cancer death worldwide. HCC progression and metastasis are closely related to altered mitochondrial metabolism, including mitochondrial stress responses, metabolic reprogramming, and mitoribosomal defects. Mitochondrial oxidative phosphorylation (OXPHOS) defects and reactive oxygen species (ROS) production are attributed to mitochondrial dysfunction. In response to oxidative stress caused by increased ROS production, misfolded or unfolded proteins can accumulate in the mitochondrial matrix, leading to initiation of the mitochondrial unfolded protein response (UPR^mt^). The mitokines FGF21 and GDF15 are upregulated during UPR^mt^ and their levels are positively correlated with liver cancer development, progression, and metastasis. In addition, mitoribosome biogenesis is important for the regulation of mitochondrial respiration, cell viability, and differentiation. Mitoribosomal defects cause OXPHOS impairment, mitochondrial dysfunction, and increased production of ROS, which are associated with HCC progression in mouse models and human HCC patients. In this paper, we focus on the role of mitochondrial metabolic signatures in the development and progression of HCC. Furthermore, we provide a comprehensive review of cell autonomous and cell non-autonomous mitochondrial stress responses during HCC progression and metastasis.

## 1. Introduction

Hepatocellular carcinoma (HCC) is one of the leading causes of cancer death worldwide. It represents the fifth-highest cause of cancer death in men and the seventh-highest in women. The causes of HCC are obesity, excess alcohol consumption, cigarette smoking, and hepatitis virus infection. The majority of these factors are preventable; however, the survival rate of HCC is still less than 20% [1].

The liver is the major metabolic organ governing whole-body energy metabolism, including glucose, fatty acid, and amino acid metabolism, and it plays a central role in nutrient homeostasis [2]. When carbohydrates are abundant, glucose is utilized as the main metabolic fuel and converted into fatty acids in the liver. Glucose and fatty acid metabolism mainly occurs in mitochondria, which are abundant in the liver [2,3]. Mitochondria are intracellular double membrane-bound structures that play multiple roles in energy metabolism and cellular homeostasis, including in the regulation of cellular respiration, oxidative phosphorylation (OXPHOS), reactive oxygen species (ROS) balance, and cell death [4].

According to data on research trends from PubMed, the number of publications on the relationship between mitochondria and HCC has substantially increased over the past few decades. These studies suggest that mitochondrial dysfunction, mitochondrial stress responses, and mitoribosomal defects induced by mitoribosomal protein aberrations in cancer cells are closely associated with tumor growth and metastasis via ROS production in the mitochondria of damaged hepatocytes, metabolic reprogramming, and mitohormetic responses [5,6].

In this review, we summarize the current knowledge on altered mitochondrial metabolism in diverse cellular and animal models of liver cancer, as well as in HCC patients. We also discuss abnormal mitochondrial proteostasis, mitohormesis, and mitoribosomal function as important mechanisms for the development and progression of HCC.

## 2. Mitochondrial Dysfunction in Hepatocellular Carcinoma

The liver is responsible for the regulation of metabolic processes, including neutralization of toxic substances, glycogen storage, and metabolism-related hormone production [2]. Based on the number and density of mitochondria in the liver, metabolic pathways in mitochondria, including β-oxidation, the tricarboxylic acid (TCA) cycle, ketogenesis, respiratory activity, and ATP synthesis through oxidative phosphorylation (OXPHOS) can be altered under physiologic and pathologic conditions [7,8]. Moreover, mitochondria have been shown to perform vital roles in the regulation of cell death signaling and innate immunity [9], and mitochondrial DNA (mtDNA) damage and abnormal OXPHOS-mediated production of ROS [10] are also associated with the development of diverse liver diseases, including liver cancer [7].

Cancer cells produce large amounts of lactate regardless of the availability of oxygen, which is referred to as aerobic glycolysis. In contrast to normal proliferating cells, most cancer cells rely on aerobic glycolysis rather than oxidative phosphorylation for glucose metabolism. This phenomenon is known as the Warburg effect. Warburg hypothesized that mitochondrial defects are developed in cancer cells; these defects lead to impaired aerobic respiration and become dependent on glycolytic metabolism [11]. In the presence of oxygen, normal cells produce NADH through the TCA cycle, and the NADH produced is used as an electron source in mitochondria to initiate electron transfer and generate ATP. By contrast, under the anaerobic conditions frequently found in tumor cells, electron flow does not happen via this process because oxygen is not available as an electron acceptor. This abnormal electron flow in the electron transport chain generates a large amount of ROS [12], which in excess cause oxidative stress, lipid peroxidation, mtDNA oxidation and mutation, induction of pre-apoptotic cytokines, and impairment of oxidative phosphorylation [13].

Mitogen-activated protein kinase (MAPK) signaling in the liver is activated by an increased ROS response. c-Jun N-terminal kinase (JNK) phosphorylation results in impaired mitochondrial function, increased oxidative stress due to blockade of electron transport and ROS production, and cell death [14,15]. AMP-activated protein kinase (AMPK) regulates cellular energy metabolism by activating mitochondrial ATP production [16]. AMPK also activates peroxisome proliferator-activated receptor gamma coactivator 1-alpha (PGC-1α), which, as a key regulator of mitochondrial metabolism and antioxidant defense, is required for mitochondrial adaptive responses to oxidative stress [17,18]. PGC-1α is a key regulatory target of sirtuin-1 (SIRT1) [19], which can act as a tumor promoter or suppressor depending on the tumor cell type [20,21]. SIRT1 is upregulated in HCC and facilitates tissue invasion and migration by activating the SIRT1/PGC-1α axis to increase mitochondrial biogenesis and ATP production [19].

In cancer cells, mtDNA alterations accumulate as a result of ROS production and abnormal mtDNA repair and lead to defects in mitochondrial respiration and ATP generation [22]. As a result of DNA mutations, the mtDNA copy number is lower in HCC than in normal tissues, which causes mitochondrial dysfunction and tumor progression [23,24]. Moreover, mtDNA mutations and copy number changes lead to defects in ATP generation through OXPHOS [24]. Mutation and deletion of mtDNA can also affect mitochondrial dysfunction and activate the mitochondrial unfolded protein response (UPR^mt^) [25]. Moreover, mitochondria transcription factor A (TFAM) is critical for activation of promoter-specific mtDNA transcription and is essential for mtDNA maintenance and stability [26]. TFAM plays essential roles in mitochondrial biogenesis and energy metabolism [27]. Expression of TFAM is downregulated by increased ROS levels, and decreased TFAM levels contribute to abnormal interactions of TFAM with mtDNA [28]. Thus, mtDNA depletion, inhibition of mtDNA transcription, and mitochondrial respiratory system defects can promote HCC progression [29,30].

Taken together, abnormal mitochondria-mediated ROS accumulation and mtDNA alterations promote HCC progression by regulating signal transduction and mitochondrial biogenesis in cancer cells.

## 3. Integrated Stress Responses, Including Cell Non-Autonomous and Autonomous Signaling

### 3.1. Cell Non-Autonomous Signaling

Proteotoxic and oxidative stress caused by increased ROS can lead to accumulation of misfolded or unfolded proteins in the mitochondrial matrix, thereby activating UPR^mt^ [31]. UPR^mt^ is regulated by the transcription factor CCAAT enhancer binding protein (C/EBP) homologous protein (CHOP), which leads to increases in the levels of mitochondrial chaperones and proteases, such as heat shock protein 60 (HSP60) and the caseinolytic mitochondrial matrix peptidase proteolytic subunit (ClpP) [32]. These mitochondrial chaperones and proteases play pivotal roles in the regulation of mitochondrial protein homeostasis [33,34]. Activating transcription factor 5 (ATF5) was identified as the mammalian ortholog of ATFS-1 and has been shown to act downstream of CHOP [35,36]. Furthermore, JUN signaling increases CHOP expression in response to the accumulation of unfolded proteins in the mitochondrial matrix, which would relieve cells from the stress [32]. UPR^mt^ is also regulated by the mitochondrial NAD-dependent sirtuin deacetylase SIRT3, and the SIRT3-UPR^mt^ axis activates forkhead box O3 (FOXO3), resulting in an increase in antioxidant functions via superoxide dismutase 2 (SOD2) activation [31,37]. Although SOD2 is overexpressed in non-invasive cells, the SIRT3-UPR^mt^ axis increases cellular invasion [37]. Thus, activation of the SIRT3-UPR^mt^ axis is positively correlated with tumor invasion and metastasis and is also associated with mitochondrial heterogeneity [38].

The fibroblast growth factor (FGF) family is widely expressed in developing and adult tissues and plays crucial roles in numerous physiological functions, including mitochondrial biogenesis, angiogenesis, cellular differentiation, and repair of tissue injury. Among FGF family members, FGF21 is abundantly expressed in the liver [39], regulates glucose, lipid, and amino acid metabolism, and is implicated in numerous metabolic diseases, including obesity, fatty liver disease, and diabetes. Hepatic expression of FGF21 is increased by peroxisome proliferator-activated receptor α/retinoid X receptor α (PPARα/RXRα) signaling during fasting and starvation, which change the levels of fatty acids [40,41,42,43]. FGF21 expression is also regulated by carbohydrate response element binding protein (ChREBP), PPAR-γ, liver X receptor (LXR), and farnesoid X receptor (FXR)/RXRα under diverse conditions [44,45,46,47]. Moreover, p53 and signal transducer and activator of transcription 3 (STAT3) regulate FGF21 expression in the liver. Under diverse types of cellular stress, including endoplasmic reticulum (ER) stress, mitochondrial stress, and oxidative stress, FGF21 expression is upregulated by p53 and STAT3. Increases in the expression of p53 and STAT3 have been found to contribute to liver damage and HCC progression by regulating FGF21 [48,49]. Thus, FGF21 can be regarded as a novel hepatokine or biomarker of the functional status of hepatocytes during liver injury. Upregulation of mammalian target of rapamycin complex 1 (mTORC1) causes glutamine depletion in the liver, in response to which FGF21 expression is upregulated by the transcriptional coactivator PGC-1α. Hepatic mTORC1 signaling regulates hepatic metabolism and whole-body physiology, and mTORC1 positively correlates with FGF21 expression in glutamine-addicted human HCC. Accordingly, blocking of mTORC1 using rapamycin can prevent glutamine depletion and FGF21 expression, which has beneficial effects on whole-body physiology and can inhibit tumor growth [50]. FGF21 can also indirectly promote beneficial effects in other metabolic organs via endocrine crosstalk. Upon liver injury, FGF21 is highly induced and attenuates or protects against hepatosteatosis, reversible and irreversible liver disease, and HCC. Conversely, lack of FGF21 expression increases interleukin-17A (IL-17A) production, insulin resistance, and inflammation by free fatty acid (FFA)-mediated induction of Toll-like 4 (TLR4) signaling in hepatocytes. Upregulation of TLR4 expression results in the accumulation of FFAs via increased lipolysis in the liver. Accumulation of FFAs and FFA intermediates, such as diglycerides and ceramides, which are lipotoxic, initiates the immune response and inflammation. Eventually these accumulations increase in the level of IL-17A and contribute to the NASH (nonalcoholic steatohepatitis)-HCC transition and HCC progression [50,51,52].

Growth differentiation factor 15 (GDF15) is a member of the transforming growth factor β (TGF-β) superfamily and has anti-inflammatory properties [51,53]. GDF15 has been identified as a UPR^mt^-associated cell non-autonomous mitokine that regulates systemic energy homeostasis and feeding behavior [53,54,55]. GDF15 is a potential diagnostic biomarker of mitochondrial myopathies [56]. It is well known that activating transcription factor 4 (ATF4) and CHOP induce GDF15 expression during mitochondrial stress and dysfunction in humans and mice [57,58].

Due to the link between mitochondrial dysfunction and liver injury, GDF15 is currently considered a biomarker of diverse liver diseases [59]. GDF15 is overexpressed in tumor tissues of HCC patients, and its expression is positively correlated with HCC progression [60]. Moreover, GDF15 overexpression enhances the proliferation and invasiveness of HCC [61]. Genetic ablation of GDF15 in animal models of liver cancer inhibits tumor formation, growth, and invasiveness [60,62]. GDF15 phosphorylates AKT also known as protein kinase B (PKB), which activates the AKT/glucose synthase kinase-3β (GSK-3β)/β-catenin signaling pathway. When AKT is phosphorylated by GDF15, GSK-3β is inhibited by phosphorylated AKT. β-catenin then translocates to the nucleus and activates the transcription of target oncogenes, such as cyclin D1, MYC proto-oncogene, bHLH transcription factor (c-Myc), matrix metalloproteinase-2 (MMP-2), and MMP-7 [63]. GDF15 is significantly increased in HCC cells treated with chemotherapy or cultured under hypoxia. GDF15 upregulation in response to chemotherapy-induced DNA damage and hypoxia involves p38 MAPK, JNK, and extracellular signal regulated kinase 1/2 (ERK1/2) activation. GDF15 then induces the production of collagen in hepatic stellate cells (HSCs), thus enhancing fibrotic processes in the liver [64].

HSCs are fibroblast-like cells that play a critical role in liver fibrosis [65]. Connective tissue growth factor (CTGF) produced in cancer cells activates HSCs and accelerates the progression of liver cancer [66]. Autophagy in HSCs leads to their activation through the upregulation of autophagy-related 7 (ATG7), thereby promoting fibrosis. Activated HSCs accelerate tumor growth and progression by releasing GDF15 in an autophagy-dependent manner [67].

Consequently, OXPHOS defects, ROS production, mitochondrial dysfunction-mediated UPR^mt^, and mitokines are implicated in HCC development and progression (Figure 1). In addition to stress signaling responses that manifest within cells as cell autonomous signaling, mitochondrial dysfunction in one cell type or tissue can result in a systemic signal, such as UPR^mt^, that affects other cells in distant parts of the organism through cell non-autonomous factors such as mitokines.

### 3.2. HCC and Cell Autonomous Mitochondrial Retrograde Signaling

Mitochondria also communicate with the nucleus through cell autonomous mitochondrial retrograde signaling [68,69,70]. These signaling pathways are locally triggered within the cells and activate specific nuclear transcription factors to accommodate to tumor conditions such as oxidative stress, OXPHOS impairments, unfolded protein accumulation, membrane potential disruption, and imbalances in ATP and metabolite levels [70] (Table 1). Mitochondrial retrograde signaling-mediated regulation of transcription has been proposed as the mechanistic link between mitochondrial defects and cancer progression.

PGC-1α and SIRT1 are well known for their involvement in the interplay between mitochondria and the nucleus in response to stress [82]. The expression of PGC-1α is lower in hepatic tumor tissues than in normal tissues from HCC patients [71]. Downregulation of PGC-1α also correlates with poor prognosis in HCC [71]. PGC-1α impairs the migration and invasion of cancer cells in HCC by promoting oxidative phosphorylation and suppressing pyruvate dehydrogenase kinase isozyme 1 (PDK1) expression [71]. In addition, PGC-1α-induced inhibition of aerobic glycolysis is mediated by WNT/β-catenin signaling-dependent PPAR-γ [71]. In contrast, Yming Li and colleagues provided data indicating that SIRT1 facilitates HCC metastasis by activating PGC-1α, which promotes mitochondrial biogenesis and energy metabolism [19]. The authors found that SIRT1 was overexpressed in various HCC cell lines and tumor specimens [19]. SIRT1 depletion in HCC cells reduced mitochondrial biogenesis and mitochondrial respiratory function. Interestingly, SIRT1 positively regulates PGC-1α expression, and increases in PGC-1α expression reversed the beneficial effect of the inhibition of SIRT1 on HCC cell invasion and metastasis by increasing the expression of the mitochondrial biogenesis-related proteins transcription factor A, mitochondrial (TFAM), cytochrome C oxidase subunit 4, isoform 1 (COXIV), and translocase of outer mitochondrial membrane 20 (TOMM20) [19].

Recently, Lee et al. showed that heat shock transcription factor 1 (HSF1) is one of the transcription factors regulated by mitochondrial ROS-induced retrograde signaling [73]. HSF1 enhances HCC cell migration and invasion by activating heat shock proteins, such as HSP70 and HSP27 [73,74]. Inhibition of HSF1 repressed AKT signaling, glycolysis, and the growth of human HCC cell lines [75].

Nuclear factor-erythroid 2 like 1 (NFE2L1), also known as NRF1, is another key transcription factor that can promote hepatoma cell invasiveness via the STAT3/NFE2L1/STX12 axis [68]. A recent study demonstrated that mitochondrial respiratory defects with low NDUFA9 (a subunit of OXPHOS complex I) expression promoted NFE2L1 expression through ROS-mediated STAT3 activation [68].

Nuclear factor erythroid 2-related factor 2 (NRF2) is a transcription factor that was demonstrated to enhance the proliferation and metastasis of HCC cells by increasing Bcl-xL and MMP-9, thereby reducing apoptosis [83]. In Hepa-1 and HepG2 cells, NRF2 increased Bcl-xL expression by binding to antioxidant response elements (AREs) in the Bcl-xL promoter [77]. Interestingly, Bcl-xL expression inhibited apoptosis, promoted drug resistance, and enhanced the survival of both HCC cell lines [77,83]. Under stress conditions, binding of the substrate adaptor p62 to Kelch-like ECH-associated protein 1 (Keap1) activates NRF2-binding AREs, leading to the expression of NRF2 target genes such as *NQO1*, *HO-1*, and *FTH1* and resulting ultimately in HCC proliferation [76].

In HepG2, mitochondrial dysfunction suppresses the expression of hypoxia-inducible factor-1α (HIF-1α), an important transcription factor that modulates cellular responses to hypoxia by both activating AMPK signaling and repressing the mTOR pathway [84]. HIF-1α has been shown to play a role in the transition from oxidative to glycolytic processes by increasing glycolytic activity through the induction of glucose transporter (GLUT1), hexokinase 2 (HK2), pyruvate dehydrogenase kinase 1 (PDK1), and lactate dehydrogenase A (LDHA). HIF-1α also decreases mitochondrial activity by inhibiting complex 1 and complex 4 activities [12]. Takashi Hamaguchi et al. also reported that HCC tissues exhibit markedly increased levels of eight glycolytic genes (*GPI*, *ALDOA*, *TPI1*, *GAPD*, *PGK*, *PGAM*, *ENO1*, and *PKM*), which can be transcriptionally activated by HIF-1α, and that these increases are related to poor survival [72]. Thus, HIF-1α suppresses ROS accumulation and reduces mitochondrial biogenesis. Additionally, HEY1 was demonstrated to be a transcriptional target of HIF-1α and to alleviate ROS production by transcriptionally repressing phosphatase and tensin homolog (PTEN)-induced kinase 1 (PINK1)-associated mitochondrial biogenesis in HCC cells [12].

Using three independent models harboring mitochondrial defects with low respiratory activity, pharmacological respiratory inhibitors, and cells with mtDNA depletion, 10 common mitochondrial defect-related genes were identified. Among them, the transcriptional regulator nuclear protein 1 (NUPR1) was shown to be regulated by mitochondrial defect-associated Ca^2+^ signaling in SNU hepatoma cells [69]. The authors of this study suggested that mitochondrial respiratory defects regulate intracellular Ca^2+^ levels to trigger the NUPR1-granulin pathway, leading to liver cancer progression [69]. Another study also indicated that NUPR1 plays a key role in controlling HCC cell growth, migration, and survival [78]. NUPR1 suppression increased the sensitivity of HCC cells to sorafenib (an effective therapeutic for HCC) and regulated a variety of important genes in HCC development such as *TGFB2*, *FGF19*, *BMP7*, and *RUNX2* [78].

Mitochondrial calcium uniporter regulator 1 (MCUR1) is upregulated in HCC cells [79]. Elevated ROS production induced MCUR1-mediated mitochondrial Ca^2+^ uptake, leading to p53 degradation, and facilitated cell survival by inhibiting mitochondria-dependent apoptosis [79]. The expression of many mitochondrial genes, including CREB, MCU, MICU1, and MICU2, has been reported to regulate mitochondrial Ca^2+^ uptake and to correlate with survival in HCC patients [80]. Specifically, a lower CREB/MICU1 ratio and a higher MCU/MICU2 were found in advanced HCC tissues [80]. Therefore, modulating mitochondrial Ca^2+^ by regulating the expression of these genes may be a potential therapeutic strategy for HCC [80]. Another study found that MCUR1 activated ROS/NRF2/Notch signaling and facilitated the epithelial-mesenchymal transition (EMT) and metastasis in HCC [81].

There is growing evidence that altered mitochondrial metabolism affects transcription in the nucleus through mitochondrial retrograde signaling, which might be one reason why mitochondrial defects contribute to tumor progression. Various types of inducers including changes in the intracellular levels of ROS, Ca^2+^, and numerous oncometabolites can trigger mitochondrial retrograde signaling during liver cancer progression. Therefore, these changes in mitochondria and mitochondrial retrograde signaling might be promising targets for anti-liver-cancer therapy.

## 4. HCC and Shifted Mitochondrial Metabolism

Metabolic reprogramming, a key feature of which is the Warburg effect, is widely recognized as a major hallmark of cancer, including HCC [71,85,86,87,88,89,90,91,92,93,94,95]. Several studies not only suggested that tumor cells depend on glycolysis as a primary energy source but also implicated mitochondrial dysfunction in the initiation and progression of HCC [85,92,93,94].

The expression of several glycolytic genes, including *HK2*, *ALDOA*, *PFKL*, *PDK1*, *GAPDH*, *PKM2*, and *PGAM1*, was significantly increased in HCC cells or tissues [92,94]. *CUEDC2*, *HMGB2*, *PFKFB4*, *PFKP*, *MACC1*, and *SIX1* were also found to be increased in advanced-stage liver cancer [86].

Several miRNAs have been shown to play a potential role in glycolysis and to be associated with HCC progression. These include miR-199a-5p, miR-517a, miR-885-5p, miR-129-5p, and miR-3662 [87,88,89,96]. While miR-517a acts as an oncogene to promote glycolysis in HCC, miR-199a-5p and miR-885-5p suppress glucose consumption, lactate production, cell proliferation, and tumorigenesis in liver cancer cells by targeting HK2 [87,88,96]. miR-3662 suppresses the Warburg effect and HCC progression by decreasing HIF-1α expression [89]. miR-129-5p is significantly suppressed in many primary HCC cell lines, and restoration of miR-129-5p expression represses glycolysis, leading to inhibition of tumor growth [97].

As described above, PGC-1α inhibits HCC metastasis by regulating the WNT/β-catenin/PDK1 axis, thereby suppressing the Warburg effect [71]. Alterations in the expression of both glycolytic enzymes and respiratory chain subunits have been proposed to contribute to HCC development. PGC-1α promotes oxidative phosphorylation while suppressing glycolysis in HCC cells [71]. Furthermore, glycolysis with consequent lactate production was found to be one of the most common metabolic changes in HCC [98]. Overexpression of PGC-1α in HCC cell lines increased ATP production and decreased extracellular lactate levels [71]. The observation of increased lactate production in NASH patients also suggests that a shift toward glycolysis occurs during HCC initiation [99].

Mitochondrial respiratory defects, as determined by the cellular oxygen consumption rate, have been observed in a variety of HCC cell lines, including Malhlavu, SK-HEP-1, and HA22T/VGH cells [100]. These HCC cell lines exhibited high glycolysis levels and were more resistant to mitochondrial inhibitors and biguanide drugs, including metformin and phenformin [100]. According to the Warburg effect, the shift from a normal cell to a cancer cell is due to impaired cell respiration. A combination of increased glycolytic flux and low cell respiration by dysfunctional mitochondria can stimulate tumorigenesis in non-cancerous cells. A study by Santacatteriba F. et al. demonstrated that inhibiting ATP production by activating the ATPase inhibitory factor 1 (IF1) can increase DEN-induced HCC in mice [101]. By contrast, cancer cell migration requires high levels of energy and mitochondrial activity, which increases oxygen consumption and ATP production. The overexpression of the ubiqinol-cytochrome C reductase hinge protein (UQCRH), a component of the respiration complex, is frequently found in advanced HCC tissues [102]. The interplay between the Wnt/β-catenin pathway and mitochondrial activity has been found to regulate the metabolic network in HCC cells [103]. Wnt/β-catenin inhibitors, including FH535 and Y3, impair mitochondrial OXPHOS and disrupt transmembrane potential and the electron transport chain, thereby reducing ATP production and inhibiting the proliferation of Huh7, Hep3B, and PLC/PRF/5 cells. Furthermore, it is reported that SIRT1 promotes HCC metastasis by activating SIRT1/PGC1-α to enhance mitochondrial biogenesis and ATP production [19]. Hence, there are differences in mitochondrial activity between tumorigenesis and metastasis in HCC.

An increasing number of studies have targeted molecules and signaling pathways that inhibit the Warburg effect in HCC. Fructose-1,6-bisphosphatase 1 (FBP1) exerted a tumor-suppressive role in HCC by inhibiting the rate of glycolysis and the glycolytic capacity of HepG2 and SK-Hep1 cells [104]. FX11 is a selective small molecule inhibitor of LDHA that can inhibit glycolysis as well as reduce lactate production. FX11 has been shown to decrease HCC cell (HepG2 and Huh7) proliferation and metastasis [104]. As a GAPDH inhibitor, koningic acid (KA) significantly reduces glycolysis and impairs the viability and migration of Hep3B, Huh-7, and Bel7407 HCC cells [86].

HK2, which catalyzes the first step in glucose metabolism, was found to be highly expressed in HCC and to promote tumor cell proliferation [105]. HK2 depletion inhibited hepatocarcinogenesis in mice, impaired HCC cell proliferation, and increased cell death through a reduction in glycolysis and a compensatory upregulation of oxidative phosphorylation [105]. Downregulation of HK2 can enhance sorafenib-induced cell growth inhibition [106]. The two most potent glycolytic inhibitors, 2-deoxy-D-glucose (2-DG) and 3-bromopyruvate (3-BP), target HK2 [107]. However, these glycolytic inhibitors still exhibited low anticancer efficacy when used alone [105]. The combination of HK2 inhibitors and an anticancer agent such as metformin or sorafenib has been shown to be an effective therapeutic for HCC [105]. Pyruvate dehydrogenase (PDH) is a mitochondrial complex that converts pyruvate to acetyl-CoA. This glycolytic enzyme controls the amount of pyruvate that enters the mitochondria for oxidative phosphorylation and lactate production. Dichloroacetate (DCA), an inhibitor of PDH kinase, has been shown to enhance the sensitivity of HCC to many chemotherapeutic agents, including sorafenib, adriamycin, 5-flurouracil, cisplatin, and doxorubicin [108,109]. Activation of PDH by PDH kinase (PDK) inhibition with DCA increased oxidative phosphorylation, reduced aerobic glycolysis, and reversed sorafenib resistance in HCC cells [108].

Indeed, targeting the glycolysis pathway has received attention as an attractive therapeutic strategy for HCC, and combination therapies have shown promising results in preclinical models.

## 5. HCC and Mitoribosomes

Mitoribosomes are macrostructures responsible for catalyzing mitochondrial protein synthesis. Mitochondria have their own ribosomes for biogenesis of the oxidative phosphorylation system. Therefore, regulation of mitoribosome biogenesis is closely related to mitochondrial respiration, cell viability, growth, and differentiation [110]. The human mitoribosome consists of small 28S subunits (MRPS) composed of about 30 mitoribosomal proteins (MRPs) and large 39S subunits (MRPL) composed of 52 MRPs [98]. Downregulation of MRPs induced mitoribosomal defects, OXPHOS impairment, and mitochondrial dysfunction, increased the recruitment of suppressive immune cells, and activated the ECM to increase the invasion capacity of HCC tumor cells [5]. TGF-β1, which is elevated in both the plasma and tumor tissues of HCC patients, was found to be associated with mitoribosomal defects [5]. ROS induced by mitoribosomal defects may play a crucial role in the immunosuppressive nature of the cancer microenvironment by activating TGF-β1 response signatures [5].

A previous study reported that the expression of mitochondrial ribosomal protein S23 (MRPS23) is altered in several cancer types [111]. MRPS23 expression was significantly unregulated in HCC samples compared with adjacent non-tumor liver tissues [111]. Patients with high MRPS23 expression had higher tumor-metastasis-node scores and significantly worse overall survival [111]. However, the detailed mechanisms through which MRPS23 contributes to HCC are unclear. The authors hypothesized that overexpression of MRPS23 could be the link explaining the association between abnormal mitochondrial function and the development of cancer [111].

Mitochondrial ribosomal protein S5 (MRPS5) is closely related to complex I function and mitochondrial respiration [112]. MRPS5 deficiency significantly reduced the NAD^+^/NADH ratio and ROS production in liver cancer stem cells [112]. Activation of the UPR^mt^ in MRPS5 knockdown liver cancer stem cells increased mitochondrial membrane potential and reduced the oxygen consumption rate [112]. In clinical samples, the expression of MRPS5 in liver cancer tissue was higher than in adjacent normal tissue [112]. In animal experiments, Zhihao Wei et al. showed that tumor size and weight in MRPS5 knockdown mice was lower than in the control group [112]. This study also indicated that deacetylated MRPS5 promotes the mitochondrial function of liver cancer stem cells and that acetylated MRPS5 translocation to the nucleus is necessary to promote glycolysis in liver cancer cells [112]. The authors suggested that the SIRT1/MRPS5 axis plays an important role in hepatic carcinogenesis by acting as a critical regulator of metabolic reprogramming in liver cancer stem cells, and that the SIRT1-high/cytoplasmic MRPS5-high profile could be an effective predictor of poor prognosis of patients with HCC [112].

Crif1 (also known as MRPL59) is essential for OXPHOS in the mitochondrial membrane [113]. A previous study showed that the expression of Crif1 was dramatically increased in a variety of human HCC cell lines, including SNU-354, SNU-368, SNU-739, HLE, and HLF cells, as well as in primary HCC tissues [113]. Furthermore, Crif1 was found to promote HCC growth and metastasis by suppressing cell apoptosis and activating ROS/NF-κB signaling [114].

SNU354 and SNU423 are two hepatoma cell lines with defective OXPHOS [98]. It has been shown that these cell lines carry a ribosomal defect that significantly reduces their expression of MRPL13 [98]. Interestingly, Lee et al. observed that a minor reduction in MRPS15 or a minor increase in MRPL38 in SNU hepatoma cells caused mitoribosomal defects due to an imbalance in subunit expression [98]. Doxycycline (DOX) and chloramphenicol (CAP) are mitochondria-specific translation inhibitors that inhibit mitoribosome activity [98]. DOX or CAP treatment significantly impaired the mitochondrial oxygen consumption rate and increased cell invasion with no changes in the cell growth rate [98]. Thus, the OXPHOS defects in HCC cell lines might be due to mitoribosomal defects. In addition, the authors demonstrated that mitoribosomal inhibition by either pharmacological agents or by siRNA-mediated knockdown of MRPL13 regulated hepatoma cell invasiveness via CLN1, the expression of which is regulated by ROS-mediated retrograde signaling [98]. Interestingly, exogenous lactate effectively suppressed OXPHOS activity by reducing MRPL13 expression [98]. Thus, mitoribosomal defects are key metabolic alterations that are triggered and sustained by microenvironmental lactate released by glycolytic hepatoma cells [98]. Lactate prevented the expression of interferon-γ (IFN)-γ by T cells, leading to tumor immune evasion and growth in liver-specific *Crif1* knock-out mice (unpublished). Thus, it is conceivable that abnormal expression of MRPs induces mitoribosomal defects and alters OXPHOS activity, oxygen consumption, the NAD^+^/NADH ratio, and ROS and lactate production, suppressing apoptosis and triggering specific pathways that promote tumor evasion.

## 6. Future Perspectives

Mitochondria are complex organelles that affect cancer development, growth, survival, and metastasis, and actively contribute to tumorigenesis by regulating mitochondrial mass and dynamics, caspase-mediated cell death, redox balance, metabolism, and signal transduction. Although the Warburg effect is related to increased glycolysis, data on other defects in cancer metabolism, such as mitochondrial dysfunction, highlight the importance of mitochondrial functions in tumor progression. In the current review, we highlighted the roles of mitochondrial cell autonomous and cell non-autonomous responses, metabolic reprogramming, and mitoribosomal defects in the development and progression of liver cancer.

Mitochondrial research in HCC has mainly focused on the inhibition of aerobic glycosis, OXPHOS-mediated ROS production, and signaling mediated by receptor tyrosine kinases, including vascular endothelial growth factor receptor 2 (VEGFR2), platelet-derived growth factor receptor (PDGFR), B-RAF, and c-kit. However, more recently, the regulation of mitochondrial proteostasis has received increasing attention, which may lead to the identification of novel druggable targets that can be targeted to improve the outcomes of HCC patients. In the near future, mitochondrial stress responses, mitochondrial one-carbon metabolism, and altered cellular energy metabolism may be attractive therapeutic targets for the treatment of HCC. Moreover, mitokines such as FGF21 and GDF15 that are induced by mitochondrial stress are invariably associated with poor prognosis in HCC patients. Mitokines are secreted by tumor-associated macrophages and cancer cells in the tumor microenvironment and are associated with aberrant growth and a poor prognosis in human HCC. Thus, mitokines in addition to mitochondrial oxidative phosphorylation could also be targeted to develop new pharmacologic agents. However, reports on mitokine-dependent effects in cancer cells are numerous, but the reported effects are complex, diverse, and inconsistent, suggesting that further investigation of the context-dependent role of mitokines in HCC development and progression is warranted.

## Figures and Tables

**Figure 1 cells-10-01901-f001:**
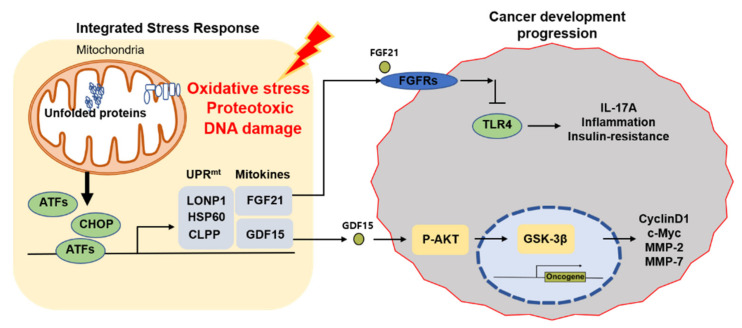
The role of UPR^mt^ and mitokines in cancer progression. When mitochondria are under stress, OXPHOS perturbations, ROS production, and the level of unfolded proteins increase. Increased expression of proteases (ClpP & LONP1) and chaperone proteins (HSP60) during UPR^mt^ promotes mitokine (FGF21 & GDF15) expression, which affects tumor development and progression. LONP1, Lon protease 1.

**Table 1 cells-10-01901-t001:** Transcriptional regulators (TRs) induced by mitochondrial retrograde signaling in HCC.

TRs	Inducers	Targets	Signaling Pathways	Ref.
PGC-1α	Oxidative stress	PDK1	WNT/β-catenin signaling	[71]
SIRT1	Oxidative stress	PGC1-α		[19]
		TFAM, COXIV, TOMM20	n.d.	
		GPI, ALDOA, TPI1, GAPD, PGK, PGAM, ENO1, PKM	n.d.	[72]
HIF-1α	Hypoxia			
		HEY1	HIF/HEY1/PINK1	[12]
HSF1	ROS	HSP70, HSP27	AKT/mTOR	[73,74]
				[75]
NFE2L1 (NRF1)	ROS NDUFA9 defects	STAT3	STAT3/NFE2L1/STX12	[68]
NRF2	Oxidative stress, p62	NQO1, HO1, FTH1	P62/Keap1/NRF2	[76]
	Oxidative stress	Bcl-xL AREs	Bcl-xL AREs	[77]
NUPR1	Ca^2+^	TGFB1, FGF19, BMP7, RUNX2	NUPR1/granulin	[69,78]
MCUR1	ROS	p53	n.d.	[79,80]
	Ca^2+^	NRF2	NRF2/Notch	[81]

n.d., not determined.

## Data Availability

Data sharing is not applicable to this Review article.

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
