# Peer review of "Mitochondrial Metabolic Signatures in Hepatocellular Carcinoma"

_cells, 2021, doi:10.3390/cells10081901_

Round 1

Reviewer 1 Report

The authors have reviewed the characteristics of HCC from the viewpoints of metabolic state and stress response in mitochondria. Scientific significance and originality of this review are splendid. To strengthen the message of this nice piece of work, I have the following concerns.

Minor

Section 1 and 2 included too much review papers: 5/6 in section 1, 12/25 in section 2, and 17/31 in section 1&2. I felt that this paper seemed to be a review of review papers at the beginning. However, the reviewer was gradually attracted by the content thereafter. Indeed, it is excellent.

Please revise section 2 by citing original papers (first report, landmark paper, and/or the latest) instead of review papers. Thank you.

Reviewer 2 Report

The Review article by Lee, H.Y. et al entitled “Mitochondrial Metabolic Signatures in Hepatocellular Carcinoma” is a comprehensive review on cell autonomous and cell non-autonomous mitochondrial stress responses during HCC progression and metastasis.

 Reviewer decision: Minor Revision Minor issues:

  • Legends for Figure 2.
  • Legends for Table 1.
  • Many abbreviations required to be written in complete name for the first time (highlighted in yellow in the chosen file), in addition to write an abbreviation list arranged in an alphabetical order.
  • The English in the present manuscript require improvement. Please carefully proof-read spell check to eliminate grammatical errors.
    1. Line 69: vital roles in the the regulation of cell death signaling
    2. Line 298: miR-199a-5p and miR-885-5p suppresses glucose consumption
    3. Line 325: diethynitrosamine
    4. Line 346: Dichloroacetate (DCA), a inhibitor of PDH kinase

Reviewer 3 Report

This review describes metabolic changes associated with hepatic carcinoma. However, in the current form, the article is very difficult to read.  The authors simply listed all those findings without adequate organization and coherence, thus making it extremely difficult to follow. At the same time, the review is inundated with alphabet soup of all these molecules, overwhelming the readers.  Considerable reorganization is necessary.

Writing a review article is not an easy task.  A well-written review article would present the progressive development of the over-arching argument.  It would begin with the basic facts or describe the current state of the knowledge, so that the nature of the issues can be understood.  Ideally, this would be followed by a description of the main issues, the supportive evidence, and the counter arguments. Finally, the authors can provide summary for what all those mean and new perspectives. 

Including an increase of publication number (Fig. 1) is odd.  Including a sentence to describe an increasing trend would be sufficient.   After all, it is a scientific paper.

Lines 76 – 78. The description does not make sense.  The metabolic hallmark of cancer cells is aerobic glycolysis, which means that cancer cells do not use oxygen (in mitochondria) even though it is available.  Although NADH is normally used for mitochondrial electron transport, aerobic glycolysis uses up NADH to generate lactate, so mitochondria would not consume oxygen.

Lines 117 – 124 seems contradictory to each other. UPR activates CHOP to upregulate chaperones and proteases in order to ease the proteotoxicity of unfolded proteins. However, right after this sentence, it states that CHOP expression accumulates unfolded proteins in mitochondria??

Cell non-autonomous vs. cell autonomous signaling is not clear.  What are the differences? Why is retrograde signaling cell autonomous but the other is not?  As described, the other is also mitochondria-to-nucleus signal.  There is no description of factors outside the cell that make cell non-autonomous.

As described, there is a differential requirement of mitochondrial energetics for tumorigenesis and cell migration for metastasis. Switching on and off of mitochondrial activity may be finely regulated for active tumor growth vs. metastasis during cancer progression.  Are there any studies on this aspect?

Round 2

Reviewer 3 Report

Lines 75 – 76. The revised sentence needs more attention.  You are talking about aerobic glycolysis of cancer cells, and, all of a sudden, mentioning tumor cells in “hypoxic/anaerobic” conditions, so cancer cells are oxygen depleted…  This may be true, but in this context, you need to describe why cancer cells do not use the available oxygen (a shift in metabolic programming to glycolysis?, stalled TCA cycle?, electron transport defect? other?).

Line 77.  What imbalance? 

Lines 120 – 122.  The authors did not understand what the reviewer was pointing out and did not make proper revision. “Furthermore, JUN signaling…., contributing to the accumulation of unfolded proteins in the mitochondrial matrix [37]”.  “UPR” is the unfolded protein response, which refers to the cellular response when unfolded proteins accumulates.  What you wrote here is that increased CHOP causes the accumulation of unfolded proteins in the matrix, which does not make sense at all.  JUN signaling upregulates CHOP IN RESPONSE TO the accumulation of unfolded proteins, which would relieve cells from the stress. Also, need to cite the original papers, not a review article.  

Author Response

Response to Reviewer 3 Comments

Point 1: Lines 75 – 76. The revised sentence needs more attention.  You are talking about aerobic glycolysis of cancer cells, and, all of a sudden, mentioning tumor cells in “hypoxic/anaerobic” conditions, so cancer cells are oxygen depleted…  This may be true, but in this context, you need to describe why cancer cells do not use the available oxygen (a shift in metabolic programming to glycolysis?, stalled TCA cycle?, electron transport defect? other?).

Response 1: We thank the reviewer for pointing this out, We edited the manuscript (line 75-76) to : “Warburg hypothesized that mitochondrial defects are developed in cancer cells, this defects lead to impaired aerobic respiration and become dependent on the glycolytic metabolism.”

Point 2: Line 77.  What imbalance? 

Response 2: We thank the reviewer for pointing this out, We edited the manuscript (line 77) to : “In the oxygen presence condition, the normal cells produced the NADH through TCA cycle and these NADH acts as electron source in mitochondria to initiate electron transfer and generate ATP. But in anaerobic conditions frequently found in the tumor cells, electron flow does not happen normally by lack of oxygen as an electron acceptor. This abnormal electron flow in the electron transport chain generates a large amount of ROS.”

Point 3: Lines 120 – 122.  The authors did not understand what the reviewer was pointing out and did not make proper revision. “Furthermore, JUN signaling…., contributing to the accumulation of unfolded proteins in the mitochondrial matrix [37]”.  “UPR” is the unfolded protein response, which refers to the cellular response when unfolded proteins accumulates.  What you wrote here is that increased CHOP causes the accumulation of unfolded proteins in the matrix, which does not make sense at all.  JUN signaling upregulates CHOP IN RESPONSE TO the accumulation of unfolded proteins, which would relieve cells from the stress. Also, need to cite the original papers, not a review article.  

Response 3: In accordance with the reviewer’s suggestion, we cited the original paper not a review article and edited the manuscript (line 120 – 122) to : JUN signaling increases CHOP expression in response to the accumulation of unfolded proteins in the mitochondrial matrix, which would relieve cells from the stress.